# Case Report: An Intracranial *Aspergillus* Infection with Cyst Formation

**DOI:** 10.3390/brainsci13020239

**Published:** 2023-01-31

**Authors:** Yu-Chun Pei, Guo-Hao Huang, Guo-Long Liu, Yan Xiang, Lin Yang, Sheng-Qing Lv, Jun Liu

**Affiliations:** Department of Neurosurgery, Xinqiao Hospital, Army Medical University, Chongqing 400037, China

**Keywords:** intracranial fungal infection, cyst formation, neurosurgery, magnetic resonance imaging

## Abstract

Intracranial fungal infection is a rare condition that often requires surgical intervention. In this study, we present a case of intracranial fungal infection with a space-occupying effect and a long medical history of five years. We comprehensively evaluated the medical history, symptoms, imaging manifestations, and pathological examinations of the patient to confirm this rare case of fungal infection with cyst formation. Moreover, we reviewed the literature on intracranial fungal infection, hoping to draw awareness and attention to this rare disease.

## 1. Introduction

Intracranial fungal infection is caused by fungus invading the meningeal or parenchymal of the central nervous system (CNS) [1]. The clinical manifestations of these infections are various and can be characterized by meningitis, encephalitis, brain abscess, space-occupying lesion, hydrocephalus, stroke, vasculitis, and spinal infection [1,2]. Fungal infections of the CNS are frequently lethal, and their diagnosis and clinical management are challenging. Robust epidemiological data on the burden of intracranial fungal infection are not readily available [3]. Globally, *Cryptococcus* spp. remains the most common cause of fungal meningitis, especially for those with uncontrolled HIV infections, with an estimated annual incidence of 2.1 to 3.9 per 100,000 population [4]. Other fungi frequently infecting the CNS include the *Candida* species, *Aspergillus*, or *Mucor* [1].

The incidence rate of this rare kind of deep fungal infection is rising because of the wide application of broad-spectrum antibiotics, corticosteroids, and immune inhibitors, the growing adoption of organ transplantation, and the increasing prevalence rate of diabetes mellitus and HIV infection [1,2]. To date, most studies on intracranial fungal infection were derived from the abovementioned conditions [5,6,7,8,9,10], whereas data on immunocompetent hosts are relatively limited [11,12,13]. In immunocompetent individuals, intracranial fungal infection can be caused by the direct inoculation of fungi through neurosurgical procedures, contaminated devices, or drug preparations, and dissemination after the inhalation or aspiration of fungal spores. In the present study, we share a rare case of an intracranial fungal lesion with a space-occupying effect from an immunocompetent patient who tolerated this lesion and depicted no obvious symptoms for almost five years. This case may show competition between the fungi infection and the immune system of the patient and present comprehensive results of this phenomenon.

## 2. Case Report

### 2.1. Clinical Presentation

A 46-year-old man was diagnosed with a right occipital lobe occupation on 21 October 2015 (Figure 1A–E) without any clinical symptoms and did not receive any medical treatment due to his own reasons. A magnetic resonance imaging (MRI) examination was performed yearly, and no change in the volume of this occupation was reported on 19 April 2018. One year later, the patient began to feel dizziness, and one month later, his clinical symptoms were aggravated. The intracranial MRI on 9 July 2020 indicated that his occupation was significantly larger than before (Figure 2A–E). Therefore, he was admitted to our hospital for surgical management.

The medical history includes a 10-year left hemifacial spasm, a cerebral infarction three years before admission, and hypertension with good blood pressure control. No history of seafood consumption, hepatitis, tuberculosis, and other chronic diseases. No history of administration of immunosuppressive agents, antibiotics, or corticosteroids. No history of exposure to environmental fungal elements. Smoking for 30 years, approximately 20 cigarettes per day. Drinking for 20 years, approximately 200 mL each occasion.

The muscle strength of the left limbs was grade V, whereas that of the right limbs was grade IV. Other physical examinations, including vital signs, consciousness, pupils, speech, and other higher cortex functions, were negative.

### 2.2. Auxiliary Check

Laboratory examination: the routine blood examination revealed white blood cell (WBC) of 11.37 × 109/L, neutrophil of 60.7%, lymphocyte of 31.1%, and eosinophil of 0.34 × 109/L. No other abnormalities in biochemistry, coagulation, C-reactive protein, urine, and stool routine were detected.

Imaging examination: the intracranial MRI revealed a space-occupying lesion with a regular shape and clear boundary in the right occipital lobe. Further, the corpus callosum and posterior horn of the right lateral ventricle were compressed. The occupation was 2.8 × 1.7 and 4.0 × 2.6 cm2 in size at the maximum imaging level on 21 October 2015 (Figure 1A–E) and 9 July 2020 (Figure 2A–E), respectively, with annular low signal shadow in the edge and small edema signal in the surrounding area, showing a high signal in the T2-weighted and T2 flair image and an equally high signal in the contrast-enhanced T1 image (Figure 1A–D and Figure 2A–D). The T1-weighted image of this occupation first revealed an equally low signal, and after the occupation became larger, it demonstrated a high signal (Figure 1A and Figure 2A). Moreover, the susceptibility-weighted imaging (SWI) examination found no vascular diseases (Figure 2E), and the CT revealed a low density of this occupation (Figure 1E). 

### 2.3. Treatment

After a complete preoperative examination and preparation, surgical resection was performed under general anesthesia on 16 July 2020. During the operation, a lesion was observed after a 4.0 cm subcortical incision in the right occipital lobe. The lesion was approximately 3.0 cm × 4.0 cm × 3.5 cm in size, with a tawny surface, clear boundary, and complete capsule (Figure 3A,B). Inside the lesion was a dark green, viscous oil-like cystic fluid. The lesion was close to the midline on the medial side, with the great cerebral vein below, and the lateral side was adjacent to the occipital angle of the lateral ventricle (for further surgical approach in detail, please see the Surgery Video in Appendix A). After surgery, the patient underwent hemostasis, antibiotics to prevent infection, local wound dressing, and other postoperative measures. It is worth noting that no antifungal therapy was adopted for this patient after surgery, as a fungal infection was confirmed after the patient was discharged from the hospital.

### 2.4. Diagnosis

The microscopic view revealed fungal spores and eosinophils in the inner side of the cyst wall, vascular proliferation and dilatation, multinucleated giant cells, eosinophils, neutrophils, foam-like tissue cells, lymphocyte infiltration, hemosiderin deposition with hemorrhage, and fibrous tissue hyperplasia with hyaline degeneration (Figure 3C,D). The immunohistochemistry results depicted that pathological markers, including S-100, and D2-40, were positive, whereas the GFAP, EMA, and PR were negative. A cellular pathological smear examination showed no tumor cells. The final pathological report showed (right occipital lobe occupation) a fungal infection with cyst formation. To further identify the potential pathogenic fungus, high throughput sequencing of the DNA (PMseq-DNA, The Beijing Genomics Institute, Beijing, China) from the pathological sections was performed. Additionally, the results show that the most-detected DNA sequences were from *Aspergillus*. After comprehensively considering the medical history, symptoms, imaging manifestations, and pathological examinations of the patient, we diagnosed this case as a rare fungal infection.

### 2.5. Outcome and Follow-Up

After 12 days of postoperative treatment, the patient recovered without complications and was discharged. No recurrence was observed in his last clinical follow-up on 1 August 2022 (Figure 4A–D).

## 3. Discussion

Intracranial fungal infections are rare and mainly occur in immunocompromised individuals. The most common pathogens of intracranial fungal infection are *Cryptococcus neoformans*, *Candida albicans*, and *Aspergillus* [1]. There are three common infection pathways. Direct invasion: *Aspergillus* often invade through the sinus, middle ear, or mastoid; blood-borne transmission: fungal septicemia caused primarily by a lung infection or intravenous drug abuse; neurosurgery, open craniocerebral trauma, lumbar puncture, penetrating injury, or drowning [1,14].

The clinical manifestations of intracranial fungal infection are various and can be characterized by acute or chronic meningitis, encephalitis, fungal brain abscesses, space-occupying lesions, hydrocephalus, stroke, vasculitis, or spinal infection [1,2]. The onset of intracranial fungal infection is often insidious and presents as a chronic or subacute process [1,14]. Initially, patients can present with a fever, headache, epilepsy, mild increased intracranial pressure symptoms, or focal neurological function damage symptoms. The increased intracranial pressure then becomes more prominent with the progress of the disease. It often takes more than four weeks from the onset of the disease to the appearance of obvious clinical manifestations; however, some patients also have an acute onset, especially in patients with severe immune deficiency [8,15]. Herein, we present a space-occupying intracranial fungal lesion in an immunocompetent patient, which persisted for five years without specific clinical manifestations; no therapeutic interventions were provided during this time as decided by the patient. Only in the last year did the patient have progressive dizziness, yet no symptoms of meningoencephalitis such as headache and fever, and no definite signs of focal neurological function were observed. This misleading manifestation hindered us from preliminarily considering the diagnosis of fungal infection.

The reliable diagnosis of intracranial fungal infection cannot be based only on symptoms and medical history. The rapid and accurate detection and identification of microbial pathogens is the key to guiding clinical timely intervention. Lumbar puncture is sometimes the gold standard in the diagnosis of intracranial infection; however, the sensitivity is very low, and only one-third to half of the cerebrospinal fluid of patients is positive [16,17]. In the present case, as we preliminarily considered it to be a neoplastic lesion or cystic mass, and the patient had been discharged when the pathological results were reported, neither lumbar puncture for routine cerebrospinal fluid examination nor further fungal culture and identification were performed. In order to identify the potential pathogenic fungus, we instead performed a high throughput sequencing of the DNA from the patient’s pathological sections, which shows that the most detected DNA sequences were from *Aspergillus*, especially from *Aspergillus nidulans*. This indicates that *Aspergillus* may be the main cause of this space-occupying cyst.

According to the literature, most *Aspergillus* infections arise through hematogenous spread from the primary sites of infection (mostly from pulmonary) or from adjacent anatomical areas, such as the paranasal sinuses [3,18]. Less frequently, intracranial *Aspergillus* infections can be caused by neurosurgical or vascular intervention or fungal endocarditis [3,18]. However, for the present case, the exact source is still unclear. When infected with *Aspergillus* in the CNS, focal lesions or brain abscess formation are the predominant presentations. Less frequently, patients present with cerebral infarction caused by septic embolism or continuous cerebral fungal invasion by granuloma formation [3,18]. However, based on the patient’s long medical history and the intraoperative features of the lesion, we believe that our case may present another rare clinical manifestation of *Aspergillus* infection.

In terms of imaging examination, distinguishing this fungal cyst from fungal granuloma and other masses, such as brain abscesses, is therefore extremely necessary [1,2,19]. According to the literature, the MR features of fungal granuloma in the brain are as follows: equal-higher signal on T1WI and equal signal on T2WI; during enhancement, the above signals are significantly enhanced, which may be nodular, multiple small rings, and large ring thick wall enhancement, indicating the existence of fungal abscess, even when the enhancement is not as obvious as that of a brain abscess, and the small ring wall can be complete, the large ring wall is thick, the inner and outer walls are not smooth, and the wall is not continuous [19,20]. In our case, the T1-weighted image of this occupation first showed an equally low signal, which became high when this occupation became larger. We speculated that this phenomenon might be due to the increased oil-like cystic fluid in this occupation. When enhanced, the lesion was almost equally enhanced, and no nodular, multiple small rings, or large ring thick walls were observed. According to the imaging examination results, a cystic mass was first considered, with a higher possibility of a tumor. However, the imaging findings could not exclude dermoid cysts or cavernous hemangioma bleeding.

The treatment of intracranial fungal infection includes antifungal therapy with commonly used drugs such as fluconazole, amphotericin b, and 5-flucytosine; surgical treatment for brain abscess, fungal granuloma, and hydrocephalus; and symptomatic support therapy, mostly used for patients with increased intracranial pressure [21,22]. In the present case, surgical treatment was the primarily considered method because of the space-occupying effect and the increased intracranial pressure symptoms. It is worth noting that no antifungal agents were adopted for this patient, and no recurrence signs were observed for two years. We speculate that this phenomenon can perhaps be ascribed to the competent immune status of the patient and our careful protection of the cyst wall intraoperatively to prevent the spread of the infection. This section may be divided into subheadings. It should provide a concise and precise description of the experimental results, their interpretation, as well as the experimental conclusions that can be drawn.

## 4. Conclusions

Fungal infections of the CNS almost always present as a surprise to the clinician. In this study, we present a rare fungal infection with cyst formation case to draw attention to this rare clinical condition. The mild symptoms, few physical signs on the patient, and nontypical imaging manifestations impeded our preoperative judgment and choices for the ideal treatment. Future efforts should be made to comprehensively summarize the characteristics of this rare disease and to explore the underlying mechanisms of its formation.

## Figures and Tables

**Figure 1 brainsci-13-00239-f001:**
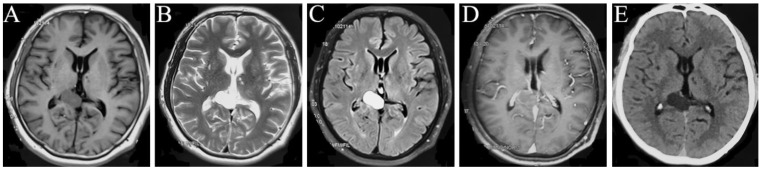
MRI manifestations of the space-occupying intracranial fungal lesion on 21 October 2015. T1WI (**A**), T2WI (**B**), T2 Flair (**C**), Contrast-enhanced T1 (**D**), and CT (**E**) show a right occipital lobe occupation, with regular shape and clear boundary, with equally high signals on T1WI (**A**), predominantly high signals on T2WI (**B**) and T2 Flair (**C**), equally enhanced signal on contrast-enhanced T1 (**D**), and low density on CT image (**E**).

**Figure 2 brainsci-13-00239-f002:**
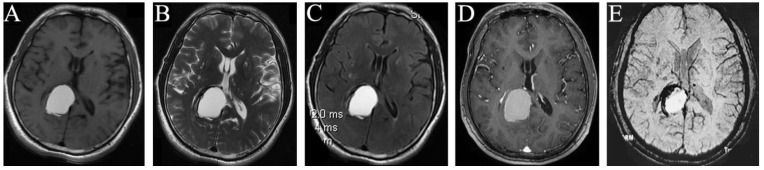
MRI manifestations of the space-occupying intracranial fungal lesion on 9 July 2020. T1WI (**A**), T2WI (**B**), T2 Flair (**C**), and Contrast-enhanced T1 (**D**) show that the right occipital lobe occupation became obviously larger than before, with predominantly high signals on T1WI (**A**), T2WI (**B**), and T2 Flair (**C**), and equally enhanced signal on Contrast-enhanced T1 (**D**); SWI examination found no vascular diseases (**E**).

**Figure 3 brainsci-13-00239-f003:**
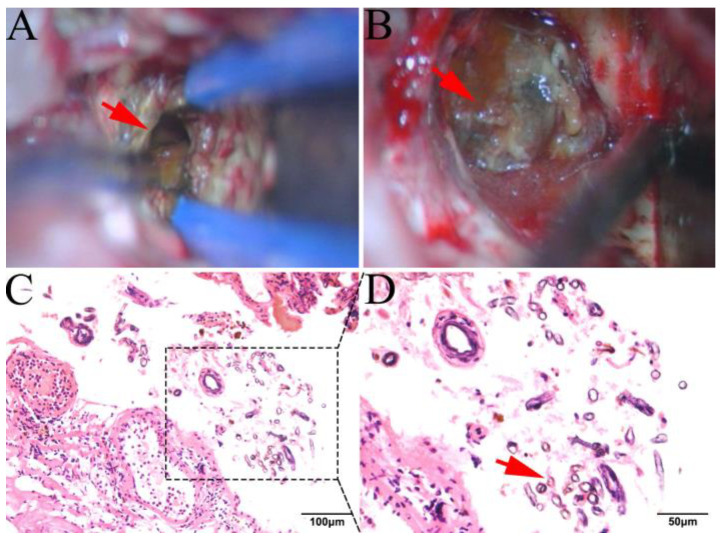
General and microscopic features of the space-occupying intracranial fungal lesion. (**A**,**B**) Intraoperative pictures showing the fungal lesion with cyst formation: (**A**) after draining out the cystic fluid; red arrow shows the cyst cavity; (**B**) before the cyst was almost totally resected; red arrow shows the cyst wall. (**C**,**D**) H&E staining results show that fungal spores and eosinophils are seen in the inner side of the cyst wall; red arrow shows the spherical and partly rod-shaped fungus.

**Figure 4 brainsci-13-00239-f004:**
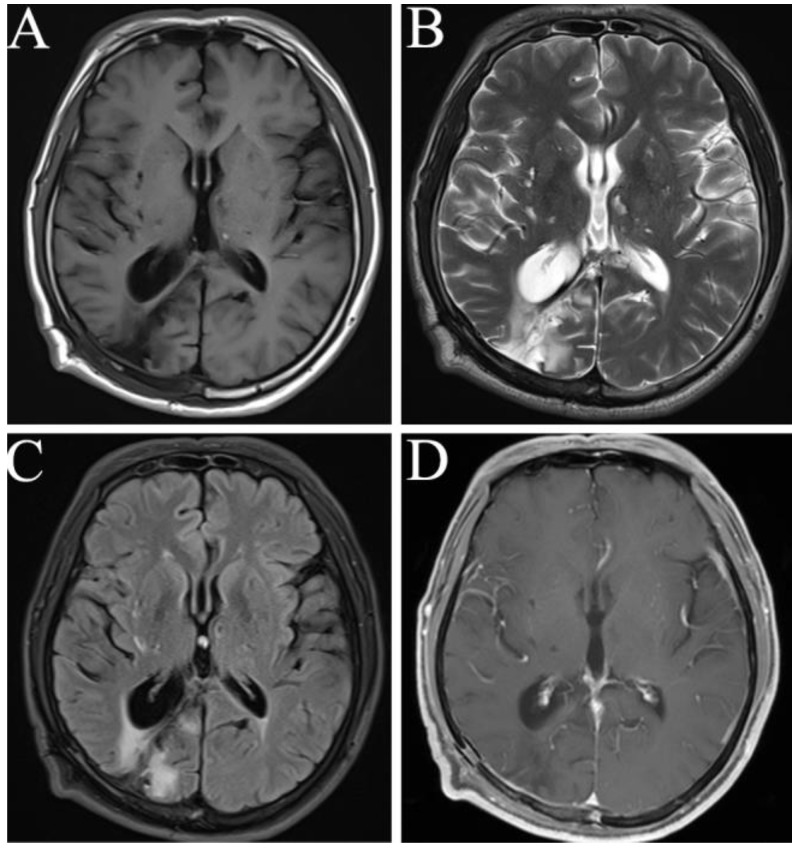
MRI manifestations in follow-up two years after surgery on 1 August 2022. T1WI (**A**), T2WI (**B**), T2 Flair (**C**), and contrast-enhanced T1 (**D**) show the postoperative changes, and no occurrence was observed.

## Data Availability

Not applicable.

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
