# Peer review of "Case Report: An Intracranial Aspergillus Infection with Cyst Formation"

_brainsci, 2023, doi:10.3390/brainsci13020239_

Round 1
Reviewer 1 Report
The presented medical case is very interesting. It may be of particular interest to clinicians. That's why I think it would be worth improving the text a bit. I propose to expand the introduction with epidemiological data or risk factors for fungal infections of the CNS. The Clinical presentation part lacks information why the patient was not diagnosed but only monitored for the first 3-4 years after finding the change. I suggest supplementing this information. Please also provide a more detailed description of the basis on which the identification was made. From the text I understand that the authors relied on the microscopic image in histological preparations. In my opinion, this is not a sufficient method.
And on a minor note: I don't understand the phrase "fungal phagocytes". Is it the phagocytic cells of the human immune system? Is it about fungi cells? There is no such term in mycology and I believe it is used incorrectly in the manuscript. Please correct. Latin names, e.g. Candida, Aspergillus, etc., should be written in italic font. Please correct. In line 81 the authors give the dimensions of the lesion as: 3.0 x 4.0 x 3.5 and add the unit as "cm3". When specifying three dimensions, the unit should be "cm". We use the unit "cm3" for volumes.
Reviewer 2 Report
The authors described a very rare case of intracranial fungal infection as a granuloma increasing in size and the treatment was successfully performed by only surgical resection. This type of intracranial Penicillium infection is very rare and has not reported in English reported literature. But some problems should be resolved before consideration of publication.
1) Title of this manuscript is inappropriate. The authors are required to change the title including Penicillium. This title is not suitable for a case report.
2) The authors should discuss about Penicillium and their patterns of intracranial infection. Otherwise, readers do not understand the significance of this case.
3) The authors need to show intracranial picture of this Penicillium mass, and to describe the surgical approach in detail.
4) Figures are a little bit to be hard to understand, therefore, figures and explanation of the pathological findings need to be correct. Especially, the point indicated with arrow in Figure A should be enlarged.
Round 2
Reviewer 2 Report
All the responses form the authors are well answered and I recognized the my questions and comments are clarified.